# New Insight on the Bioactivity of *Solanum aethiopicum* Linn. Growing in Basilicata Region (Italy): Phytochemical Characterization, Liposomal Incorporation, and Antioxidant Effects

**DOI:** 10.3390/pharmaceutics14061168

**Published:** 2022-05-30

**Authors:** Immacolata Faraone, Ludovica Lela, Maria Ponticelli, Domenico Gorgoglione, Filomena De Biasio, Patricia Valentão, Paula B. Andrade, Antonio Vassallo, Carla Caddeo, Roberto Falabella, Angela Ostuni, Luigi Milella

**Affiliations:** 1Department of Science, University of Basilicata, Viale Dell’ateneo Lucano 10, 85100 Potenza, Italy; immacolata.faraone@unibas.it (I.F.); ludovica.lela@unibas.it (L.L.); antonio.vassallo@unibas.it (A.V.); angela.ostuni@unibas.it (A.O.); 2Spinoff BioActiPlant s.r.l., Viale Dell’ateneo Lucano 10, 85100 Potenza, Italy; 3EVRA S.r.l., Località Galdo, 85044 Lauria, PZ, Italy; cto@evraitalia.it (D.G.); f.debiasio@evraitalia.it (F.D.B.); 4REQUIMTE/LAQV, Laboratório de Farmacognosia, Departamento de Química, Faculdade de Farmácia, Universidade do Porto, R. Jorge Viterbo Ferreira, n° 228, 4050-313 Porto, Portugal; valentao@ff.up.pt (P.V.); pandrade@ff.up.pt (P.B.A.); 5Spinoff TNcKILLERS s.r.l., Viale dell’Ateneo Lucano 10, 85100 Potenza, Italy; 6Department of “Scienze Della Vita e Dell’ambiente, Sezione di Scienze del Farmaco”, University of Cagliari, Via Ospedale 72, 09124 Cagliari, Italy; caddeoc@unica.it; 7Urology Unit, San Carlo Hospital, Via Potito Petrone, 85100 Potenza, Italy; rfalabella@libero.it

**Keywords:** *Solanum aethiopicum* Linn., scarlet Lucanian eggplant, antioxidant activity, HepG2, antioxidant enzymes, Nrf2 pathway, liposomes

## Abstract

Food extract’s biological effect and its improvement using nanotechnologies is one of the challenges of the last and the future decades; for this reason, the antioxidant effect of scarlet eggplant extract liposomal incorporation was investigated. Scarlet eggplant (*Solanum aethiopicum* L.) is a member of the Solanaceae family, and it is one of the most consumed vegetables in tropical Africa and south of Italy. This study investigated the antioxidant activity and the phytochemical composition of *S. aethiopicum* grown in the Basilicata Region for the first time. The whole fruit, peel, and pulp were subjected to ethanolic exhaustive maceration extraction, and all extracts were investigated. The HPLC-DAD analysis revealed the presence of ten phenolic compounds, including hydroxycinnamic acids, flavanones, flavanols, and four carotenoids (one xanthophyll and three carotenes). The peel extract was the most promising, active, and the richest in specialized metabolites; hence, it was tested on HepG2 cell lines and incorporated into liposomes. The nanoincorporation enhanced the peel extract’s antioxidant activity, resulting in a reduction of the concentration used. Furthermore, the extract improved the expression of endogenous antioxidants, such as ABCG2, CAT, and NQO1, presumably through the Nrf2 pathway.

## 1. Introduction

In the last year, the concept of the prevention of oxidative stress using natural resources has been paid particular attention, since it plays an active role in preventing several common illnesses, such as diabetes, preeclampsia, acute renal failure, high blood pressure, atherosclerosis, and neurodegeneration. Reactive species of oxygen (ROS) are normally generated by cells during oxygen metabolism, and, under physiological conditions, the rate and the amount of oxidant formation are balanced by the frequency of their remotion. However, when the balance between the antioxidant and pro-oxidant systems is lost, oxidative stress occurs [1]. Several investigations have highlighted the important role of natural antioxidants in preserving human health. For this reason, the evaluation of food antioxidant activity is crucial to understanding their potential application in preventing and treating oxidative stress-related diseases. Plants and vegetables are indeed known to possess a high content of compounds that can capture and neutralize free radicals like carotenoids, polyphenols, anthocyanins, etc., making them usable in prophylactic and curative phytotherapy. Based on this background, the antioxidant activity of a typical vegetable from the Basilicata Region (Italy), *Solanum aethiopicum* Linn., has been studied.

The scarlet eggplant, or *Solanum aethiopicum* L., is part of a widespread genus belonging to the Solanaceae family and is one of the most commonly consumed vegetables in tropical Africa. Together with tomato, onion, pepper, and okra, it is one of the five most important vegetables in Central and West Africa, but it is also cultivated in the Caribbean and Brazil. The species was known to be domesticated by the wild *Solanum anguivi* Lam. through the semi-domesticated *Solanum distichum* Schumach. & Thonn., which are both found throughout tropical Africa. The close relatives of *S. aethiopicum* are other African species, the *Solanum macrocarpon* L. and *melongena* L. [2]. In Italy, the scarlet eggplant population was only found in Basilicata in an area near the Regional Natural Reserve of Pollino, where it was utilized for many years with the brinjal eggplant (*S. melongena*) [3]. The history of the Lucanian species of *S. aethiopicum* started at the beginning of the last century, when some Lucanian emigrants in Africa decided to return to their native land carrying the seeds of this splendid plant. With the complicity of the World War, the already poor people of the area found an essential food in this particular eggplant, a rustic plant that produces red fruits that are similar to tomatoes but have a spicy, slightly bitter taste. Nowadays, this species has taken part in the Slow Food Foundation catalogue and entered the Protected Designation of Origin category (DOP) [4].

Several investigations have reported the African eggplant’s biological activity, including hepatoprotective, anti-inflammatory, nephroprotective, anticancer, antidiabetic, and antioxidant activity [5]. These healthy properties have been correlated to its phytochemical profile, since *S. aethiopicum* has been shown to be rich in phenolic compounds and alkaloids [6,7,8]. Flavonoids and ascorbic acid have been found in either the fruits or the stalks of the plant, which have shown antioxidant activity. All parts of the plant are generally used, including ash or powder, to treat diseases such as diabetes, cholera, otitis, bronchitis, toothache, dysuria, haemorrhoids, dysentery, asthenia, and skin infections in decoction [2]. These health effects are related to eggplants’ phytochemical composition, since they are rich in alkaloids, such as solasodine, and phenolic acids [2]. Specifically, eggplants, compared with the other members of the *Solanaceae,* are considered the major sources of phenolic acids, among which chlorogenic acid and its isomers (cryptochlorogenic acid, neochlorogenic acid, and *cis*-chlorogenic acid) are the most representative. Other bioactive compounds found in eggplant are flavonoids such as rutin, kaempferol, or quercetin, which have proven antioxidant and anti-inflammatory activities [6]. In contrast, the carotenoid content in eggplants is lower than that of other vegetables such as tomatoes or carrots, and among them, lutein or *β*-carotene are known to have antioxidant, anti-inflammatory, and anti-atherosclerotic activities [2,6]. However, it is worth noting that many factors influence the fruits’ and herbs’ chemical composition and thus their biological activity; among them are the geographical location, season, climatic conditions, and growing conditions [6,9]. No data are available to date on the chemical profile and antioxidant activity of *S. aethiopicum* from the Basilicata Region; for this reason, in this study, the Italian eggplant was investigated starting from extraction. Specifically, the pulp, peel, and whole fruit of the Lucanian *S. aethiopicum*, were extracted using absolute ethanol as a solvent and exhaustive maceration as an extractive technique to guarantee the extraction of the phenolic compound and carotenoids. However, despite their beneficial properties, polyphenols have low bioavailability, weak chemical stability, and poor solubility, making them difficult to translate into in vivo applications [10]. These problems can be overcome by incorporating plant extracts or the isolated polyphenols into micro/nanocarriers, which improves bioavailability and efficacy, leading to an optimal concentration at the target site [11,12]. Hence, this study aimed to investigate the phytochemical composition of the Lucanian *S. aethiopicum* (peel, pulp, and whole fruit) ethanolic extract and its antioxidant activity. Further, the extract with the best activity was incorporated into liposomes, and the increase in the antioxidant potential was investigated in vitro on cell lines.

## 2. Materials and Methods

### 2.1. Chemicals and Reagents

Absolute ethanol, dimethyl sulfoxide (DMSO), Dulbecco’s Modified Eagle Medium (DMEM), Folin–Ciocalteu reagent, sodium carbonate (Na_2_CO_3_), [3-(4,5-dimethyl-2-thiazolyl)-2,5-diphenyl-2H-tetrazolium bromide] (MTT) and 2′,7′-dichlorodihydrofluorescein diacetate (DCFH-DA), 2,2-diphenyl-1-picrylhydrazyl (DPPH), sodium acetate anhydrous, ferric chloride hexahydrate (FeCl_3_·6H_2_O), 2,4,6-tripyridyl-s-triazine (TPTZ), fluorescein, sodium phosphate monobasic (NaH_2_PO_4_), 2,2′-Azobis (2-amidinopropane) dihydrochloride (AAPH), 6-hydroxy-2,5,7,8-tetramethylchroman-2-carboxylic (Trolox), quercetin, were acquired from Sigma Aldrich S.p.A. (Milan, Italy). Fetal bovine serum (FBS), trypsin-EDTA solution, glutamine, phosphate saline buffer (PBS), and penicillin-streptomycin were taken over Euroclone (Milan, Italy). RT-PCR Reagents were purchased from Euroclone (Milan, Italy). Phospholipon 90G (>90% phosphatidylcholine; P90G) was acquired from Lipoid GmbH (Ludwigshafen, Germany). Standards of 4-*O*-caffeoylquinic acid and 4,5-dicaffeoylquinic acid were obtained from Chengdu Biopurify Phytochemicals Ltd. (Chengdu, Sichuan, China). 5-*O*-Caffeoylquinic acid, eriodictyol, eriodictyol 7-*O*-glucoside, naringenin 7-*O*-glucoside, and kaempferol 3-*O*-rutinoside were purchased from Extrasynthase (Genay, France). Quercetin 3-*O*-rutinoside (rutin), kaempferol 3-*O*-glucoside, naringenin, lutein, *β*-carotene, were purchased from Sigma-Aldrich (St. Louis, MO, USA). *α*-carotene and lycopene were purchased from Carote Nature (Lupsinggen, Switzerland). Phosphoric acid (H_3_PO_4_), ammonium sulphate (NH_4_)_2_SO_4,_ and HPLC grade methanol were obtained from Merck (Darmstadt, Germany), and formic acid was supplied from BDH Prolab (Dublin, Ireland).

### 2.2. Plant Material and Extraction

The fruits of *S. aethiopicum*, investigated in this study, were grown in Basilicata and provided by EVRA ITALIA s.r.l. (Località Galdo, Lauria PZ). The whole fruit, peel, and pulp were cut into small pieces and extracted using the exhaustive maceration technique. The plant material (peel 234 g, pulp 219 g, and entire fruit 124 g) was placed in dark bottles with absolute ethanol (plant material:solvent ratio 1:20) in the dark at room temperature for 48 h. The extraction procedure was repeated three times. The obtained extracts were filtered with filter paper, and the solvent was evaporated at reduced pressure through a rotary evaporator. The extracts were stored dried in the darkness at ambient temperature until usage.

### 2.3. Total Phenolic Content (TPC)

TPC was determined by the Folin–Ciocalteu method [13]. Briefly, 75 μL of the diluted extract was mixed with distilled water (425 μL), the Folin–Ciocalteau reagent (500 μL), and Na_2_CO_3_ (10% *w/v*) (500 μL). The obtained solution was mixed and kept in the dark for 1 h at room temperature. After this time, the absorbance of the mixture was determined at 723 nm by using a UV-Vis spectrophotometer (SPECTROstarNano, BMG Labtech, Ortenberg, Germany). TPC was reported as mg of Gallic Acid Equivalent (GAE)/g of dried extract (DW). TPC for all extracts was carried out in triplicate.

### 2.4. DPPH Free Radical Scavenging Test

The radical-scavenging ability of samples was evaluated by in vitro DPPH neutral radical, as reported by Faraone et al., 2019 [14]. Briefly, 50 μL of different extract concentrations or Trolox, used as standard, were added to 200 μL of a DPPH methanol solution (0.0476 mg/mL) in a 96-well plate. The plate was incubated in the dark and at room temperature for 30 min. After this time interval, the reaction was observed at 515 nm and results were reported as milligram Trolox equivalents per gram of dried extract (mgTE/g DW) and as concentration (mg/mL) which produces a 50% radical scavenging activity (IC_50_). DPPH assay was performed in triplicate.

### 2.5. Ferric Reducing Antioxidant Power (FRAP)

The FRAP assay was carried out as described by Vassallo et al., 2020 [12]. The FRAP reagent was prepared by mixing 38 mM sodium acetate anhydrous buffer in distilled water, pH 3.6, with 20 mM FeCl_3_·6H_2_O in distilled water and 10 mM TPTZ (2,4,6-tripyridyl-s-triazine) in 40 mM HCl (10:1:1). 180 μL of FRAP reagent and 20 μL of each extract dilution were mixed in a 96-well plate and incubated for 40 min at 37 °C in the darkness. For blank, 180 μL FRAP reagent was mixed with 20 μL methanol. The solution absorbance was measured at 593 nm. Trolox was used as a reference antioxidant standard, and results were expressed as milligrams of Trolox equivalents per gram of dried extract (mg TE/g DW). FRAP assay was made in triplicate for all samples.

### 2.6. Oxygen Radical Absorbance Capacity (ORAC) Assay

According to Sinisgalli et al. 2020 [15], 125 μL of fluorescein (10 nM in 75 mM NaH_2_PO_4_ buffer at pH 7.4) and 25 μL of the extract at different concentrations were incubated in triplicate in a 96-well microplate at 37 °C for 30 min. Subsequently, to each well was added 25 μL of 10 mM AAPH, and fluorescence (λex 485 nm and λem 520 nm) was monitored for 90 min every 2 min through a GLOMAX Multidetection System (Promega, Madison, WI, USA). As the reference standard was used Trolox (0–100 μM). Data were estimated based on the changes in the areas under the fluorescence decay curve among the blank, standards, and samples. Final ORAC values were reported as μmol of Trolox equivalents (TE)/kg of dried extract (DW).

### 2.7. HPLC-DAD Characterization

All extracts of the Lucanian species of *S. aethiopicum* tissue were solubilized in HPLC grade methanol and filtered through a 0.45 μm size pore membrane before analysis.

#### 2.7.1. Phenolic Profile

Phenols analysis was performed following the procedure described by Oliveira et al., 2009 [16]. All extracts were analysed in a volume of 20 μL using a Gilson HPLC-DAD unit and the Spherisorb ODS2 (25.0 × 0.46 cm, 5 μm particle size; Waters, Milford, MA, USA) as column which was kept at 26 °C. The mobile phase solvents were 5% (*v*/*v*) formic acid in water (phase A) and methanol (phase B) used with the following gradient programme: 0–3 min 5–15% B, 3–13 min 15–25% B, 13–25 min 25–30% B, 25–35 min 30–35% B, 35–39 min 35–45% B, 39–42 min 45% B, 42–44 min 45–50% B, 44–48 min 50–55% B, 48–51 min 55–70% B, 51–57 min 70–75% B, 57–61 min 75–80% B. The flow rate used was 0.9 mL/min. The detection was achieved with the Agilent 1100 series diode array detector (DAD) (Agilent Technologies, Waldbronn, Germany) and chromatograms were analyzed at 280, 320, and 350 nm. The purity of peaks was evaluated using the Clarity Software, version 5.04.158 (DataApex Ltd., Prague, Czech Republic). The quantification of phenolic compounds present in the extracts of the Lucanian *S. aethiopicum* fruit tissues was performed using calibration curves of the respective authentic standard analyzed under the same conditions. Flavanones (eriodictyiol-7-*O*-glucoside, eriodyctiol, and naringenin) were quantified at 280 nm, hydroxycinnamic acids (4-*O*-caffeoylquinic, chlorogenic, and 4,5-di-*O*-caffeoylquinic acids) were quantified at 320 nm and flavanols (quercetin-3-*O*-rutinoside, kaempferol-3-*O*-glucoside, and kaempferol-3-*O*-rutinoside) at 350 nm. Each extract was injected in triplicate.

#### 2.7.2. Carotenoids Profile

Carotenoids were analyzed following the procedure described by Amaro et al., 2015 [17] by comparing their UV-Vis spectra and retention times with the calibration curves of the corresponding standards. Each extract was injected in triplicate and quantified at 450 nm.

### 2.8. Liposome Preparation and Characterization

For the preparation of *S. aethiopicum* extract liposomes, 90 mg/mL of P90G and 2 mg/mL of *S. aethiopicum* peel extract were dispersed in water and subjected to sonication (23 alternate cycles of 5 s on/2 s off + 5 alternate cycles of 3 s on/2 s off; 13 μm of probe amplitude) using an ultrasonic disintegrator (Soniprep 150, MSE Crowley, London, UK). Empty liposomes were prepared following the above procedure but without including the extract.

The mean diameter, the polydispersity index, and the zeta potential of the liposomes were determined via dynamic and electrophoretic light scattering using a Zetasizer nano-ZS (Malvern Panalytical, Worcestershire, UK). The liposomes (*n* = 10) were diluted with ultrapure water (1:100 *v*/*v*) prior to the analyses at 25 °C.

### 2.9. Cell Culture and Treatment with Extracts

Human hepatocellular carcinoma cells (HepG2) were cultured in DMEM (supplemented with 2 mM glutamine, 10% fetal bovine serum, 100 μg/mL streptomycin, and 100 U/mL penicillin), and maintained at 37 °C in a humidified atmosphere containing 5% CO_2_. The *S. aethiopicum* peel extract was dissolved in DMSO, and different concentrations were tested (1–400 μg/mL). In all the experiments, DMSO-treated cells were used as the control (CTRL).

### 2.10. Cell Viability Assay

Cell viability was tested on HepG2 cells using the MTT (3-[4,5-dimethylthiazol-2-yl]-2,5 diphenyl tetrazolium bromide) assay. HepG2 cells were cultivated in a 96-well plate (1.5 × 10^3^ cells/well), incubated during the night, and treated with the extract at various concentrations (25–400 μg/mL) for 24 and 48 h. After medium remotion, PBS was used to wash cells which were subsequently incubated for 4 h with MTT solution (0.75 mg/mL) in PBS. At the end of the 4 h, the solution was removed, and a solubilization solution (1:1 DMSO:isopropanol) was used to lysed cells. The solubilized formazan product was quantified at 560 nm through the use of a UV–Vis spectrophotometer (SPECTROstarNano BMG Labtech, Ortenberg, Germany). Each extract was tested in triplicate.

### 2.11. Measurement of Intracellular ROS

Reactive oxygen species (ROS) were detected with a fluorescent probe DCFH-DA, as described by Sinisgalli et al., 2020 [15]. HepG2 cells were seeded (1.5 × 10^5^ cells/well) in a 24-well plate and treated with different doses of *S. aethiopicum* peel extract (200–1 μg/mL), liposomes or acetyl-l-cysteine (NAC) (10 mM). After 24 h, cells were stressed for 1 h with 5 mM of *tert*-butilhydroperoxide (t-BuOOH). Cells were then stained with 10 μM DCFH-DA at 37 °C for 30 min in the darkness, and fluorescence was measured by using a GloMaxMultiDetection System (Promega, Madison, WI, USA) equipped with a blue filter (ex.:490 nm; em.:510–570 nm). Each extract was analyzed in triplicate

### 2.12. Quantitative RT-PCR

HepG2 cells were treated for 24 h with the Lucanian *S. aethiopicum* peel extract at different concentrations (200–1 μg/mL). The quantitative RT-PCR was performed following the method of Armentano et al., 2018 [18] in triplicate.

### 2.13. Statistical Analysis

Data were shown as mean ± standard deviation (Mean ± SD). GraphPad Prism 5 Software, Inc. (San Diego, CA, USA) was used to carry out the statistical analysis, and *p* values ≤ 0.05 were considered statistically significant.

## 3. Results and Discussion

### 3.1. Extraction Yields and Total Phenolic Content

The most frequent technique used to isolate antioxidant compounds from plant materials is solvent extraction. The extract yields and the related antioxidant activity are strongly related to the extractive solvent nature due to a wide variety of molecules with different polarities existing in plants. Because of their polar nature, ethanol end methanol are extensively used to isolate antioxidant molecules such as phenols from vegetal materials [19]. Compared to methanol, ethanol is non-toxic to humans and usable for food and natural medicinal scopes, and either absolute or aqueous ethanol has been successfully used for phenolic antioxidant molecule extraction from vegetables with good results [19,20]. For this reason, the whole fruit, pulp, and peel of Italian *S. aethiopicum* growing in the Basilicata Region were subjected to exhaustive maceration extraction for 48 h using absolute ethanol as extractive solvent. At the end of the extraction, the solvent was evaporated, and extraction yields were calculated. The peel and the whole fruit extract showed a higher percent yield, while pulp was lower (5.82%, 5.65%, and 4.72%, respectively; Table 1).

Total phenolic content (TPC) was evaluated spectrophotometrically through the Folin–Ciocalteu method to make a first qualitative screening of extracts (Table 1). The peel extract reported the highest value of phenols (20.94 ± 0.83 mgGAE/g DW), followed by whole fruit and pulp (9.41 ± 0.17 and 6.38 ± 0.19 mgGAE/g DW, respectively). These amounts of phenolic compounds are significantly lower than that obtained from the African *S. aethiopicum* fruit by Nwanna et al. [5] (253.1 ± 7.3 and 499.2 ± 11.1 mgGAE/g) but are higher than that of Khatoon et al. [21], who evaluated the TPC on the whole fruit of Indian *S. aethiopicum*, reaching a value of 3.89 ± 0.07 mgGAE/g DW. Further, from comparison with the edible parts of other eggplant species, such as *S. torvum* and *S. macrocarpon*, it was seen that they have a lower TPC (2.03 ± 0.04 and 1.03 ± 0.05 mgGAE/g DW, respectively) than the Lucanian eggplant [21]. Finally, different African *S. auguivi* Lam fruits demonstrated TPC values near the data obtained in this investigation (from 8.04 ± 0.28 to 11.13 ± 0.18 mgGAE/g DW) [22]. The discrepancy observed may be related to the environmental differences that may influence the concentration and bioavailability of certain active molecules in herbs and vegetables. Researchers have indeed demonstrated that season, breeding medium, or even the presence of insects or animals may interfere with the presence or absence of specialized metabolites in the same plant species [5,9].

### 3.2. Antioxidant Activity

It is well known that a single test cannot fully assess the antioxidant activity of an extract, since antioxidants are a composite class of substances, and each of them, depending on their chemical structures, can counteract a specific category of radical and/or oxidizer [23]. Therefore, more than one technique is needed to understand a plant’s antioxidant characteristics. For this reason, the extracts’ antioxidant activity was evaluated through three different spectrophotometric assays: DPPH, ORAC, and FRAP (Table 1).

As can be seen in Table 1, all extracts proved to have antioxidant activity; in particular, the peel extract reported the highest radical scavenging activity, evaluated with the DPPH test (16.96 ± 0.57 mgTE/g DW; IC_50_ = 2.56 ± 0.09 mg/mL), and ferric reducing power determined using the FRAP test (21.52 ± 0.58 mgTE/g DW). Data from the DPPH assay confirm what is present in the literature, since Nwanna et al., in two different studies on African *S. aethiopicum* fruits, described a radical scavenging activity with an IC_50_ ranging between 3.23 ± 0.06 and 4.23 ± 0.14 mg/mL [5,24]. In contrast, the reducing power evaluated by Khatoon et al. on Indian *S. aethiopicum* fruits was higher than that obtained from the whole fruit of the Lucanian species (20.03 ± 0.17 mgGAE/g DW vs. 9.80 ± 0.58 mgTE/g DW, respectively) investigated in this study [21]. However, this may be due to the different antioxidant standards used for the quantification, since they used gallic acid, which is more potent than Tolox [25].

Furthermore, in the ORAC assay, all three extracts quenched peroxyl radicals produced by AAPH in a dose-dependent manner. This method is unique, since its ROS generator, the AAPH, is involved in the production of peroxyl free radicals under thermal decomposition and is also commonly found in the body, making this method more relevant from a biological point of view than DPPH and FRAP assays. Moreover, AAPH is reactive with lipid- and water-soluble substances, so it may determine a measure of total antioxidant potential and is widely used for determining the antioxidant capability of food matrices. The ORAC test measures a fluorescent signal from a probe, which is switched off in the presence of ROS; hence, the persistence of the fluorescent signal is an index of the presence of antioxidant molecules in the sample tested. Different fluorescence decay curves were obtained for different extracts (Figure 1).

As shown in Figure 1B, the pulp extract curve was steeper than that of the Trolox used as standard and the other two extracts, meaning that the fluorescence is lowered faster over time due to the greater quantity of radicals present. In contrast, the peel extract (Figure 1C) strongly slowed fluorescein degradation and reported the highest ORAC value (477.22 ± 330.64 μmolTE/kg DW), since it was three times and six times higher than whole fruit and pulp, respectively (Table 1). These results obtained from the antioxidant assays reflect the phenolic content of extracts, since the peel extract had both a greater phenolic amount and higher antioxidant activity. This is in line with the nature of phenols, since they have been regarded as potent antioxidants in vitro and have been shown to be more effective antioxidants than vitamin C, vitamin E, and carotenoids [26,27]. Specifically, phenolics’ antioxidant activity is related to their ability to donate hydrogen or electron and delocalize the mismatched electron within the aromatic structure [26].

To further validate and compare the antioxidant activity results obtained from FRAP, DPPH, and ORAC assays, the Relative Antioxidant Capacity Index (RACI) was calculated [14]. RACI is a statistical, adimensional index calculated using Excel software (2010, Microsoft, Redmond, WA, USA) by integrating the antioxidant capability values generated from the different methods used. The final value of RACI represents the standard scores transformed from the initial data generated by different assays for each extract [28]. The TPC method was also included in the RACI calculation, since it was recently proposed to determine samples’ total reducing ability, reflecting the capability of both phenolic and non-phenolic compounds to reduce the Folin–Ciocalteu reagent. In agreement with the above results, the peel extract had the highest RACI value (0.73), followed by whole fruit and pulp (Figure 2).

These data agree with other studies showing that the peel has a higher antioxidant activity than other parts of vegetables or fruits, which can be related to its protective role [29].

### 3.3. Phytochemical Characterization by HPLC-DAD

DAD enables each compound’s UV/Vis spectrum to be recorded and allocates each chromatogram peak to a specific class of metabolite depending on the absorbance maximum and the typical spectrum due to the chromophores in their molecular structure. Specifically, HPLC-DAD provided the phytochemical profile of *S. aethiopicum* tissues from the Basilicata Region; all phenolics and carotenoids were identified and quantified based on the respective standards.

#### 3.3.1. Phenolic Profile

Extracts of different Lucanian *S. aethiopicum* tissues (whole fruit, peel, and pulp) were analyzed to define their phenolic content. The HPLC-DAD analysis revealed the presence of ten phenolic compounds, including three hydroxycinnamic acids (**1**, **2**, **7**), four flavanones (**3**, **4**, **5**, **10**), and three flavanols (**6**, **8**, **9**) (Table 2).

As far as we know, only 5-*O*-Caffeoylquinic acid (**2**) and quercetin-3-*O*-rutinoside (**6**) were recently reported in African *S. aethiopicum* fruit’s aqueous and methanolic extracts [5,24,30]; the other identified eight compounds had not been listed in previous analyses. Therefore, this could be the first documented report about the presence of these compounds in *S. aethiopicum* from the Basilicata Region. In addition to these ten phenols, results showed three other compounds whose identification was not achieved, but, based on their flavanone-like UV-vis spectra, they were labeled as unknown flavanones (**a**–**c**) (Figure 3).

The total phenolic amounts ranged between 1331.44 µg/g for pulp extract and 16,126.40 µg/g for peel extract. The whole fruit and the peel extracts showed a similar chemical composition, so the variety of phenols associated with the whole fruit should be due to the peel presence. This hypothesis is supported by the fact that only three compounds were found in the pulp extract. Despite the similar qualitative phenolic profile, the overall concentration of these compounds varied between whole fruit and peel extracts. As reported in the literature, 5-*O*-caffeoylquinic acid, also known as chlorogenic acid, is the major phenolic acid found in African *S. aethiopicum* fruit [5,31]. The peel extract presented a similar amount of 5-*O*-caffeoilquinic acid (**2**) to the whole fruit, but naringenin (**10**), quercetin-3-*O*-rutinoside (**6**), and kaempferol-3-*O*-rutinoside (**9**) were the dominant phenolic compounds (Figure 3) found in this tissue. Overall, the greatest number of phenols, found in *S. aethiopicum* peel, confirms previously discussed data obtained from the spectrophotometric assays.

#### 3.3.2. Carotenoids Profile

Carotenoids are among the most important natural pigments; more than 600 distinct molecules have been identified, with *β*-carotene being the most abundant. They play an essential role in protecting plants against the photooxidative process, since they are involved in scavenging peroxyl radicals and singlet molecular oxygen. Likewise, carotenoids are involved in the antioxidant defence system of the human body [32]. Up to now, only one study has reported carotenoids analysis of African *S. aethiopicum* leaves [33]. Hence, this is the first investigation reporting the chromatographic quantification of carotenoids in eggplant’s fruit, peel, and pulp (Table 3).

Four carotenoids, including one xanthophyll (**11**) and three carotenes (**12**–**14**), were identified in whole fruit and peel ethanolic extracts. Even in this instance, the whole fruit carotenoid content that was obtained is due to the peel; for this reason, it was not quantified (Figure 4).

As expected, *β*-carotene is the most abundant carotenoid found in the peel, and the other carotenoids are among the same as those found in African eggplant leaves by Mibei et al. [33].

### 3.4. Liposome Preparation and Characterization

The higher amounts of phenolic compounds and carotenoids found in the peel extract confirm data obtained from the antioxidant assay, making this edible part of the Lucanian *S. aethiopicum* a promising source of bioactive molecules. For this reason, the peel extract was incorporated into liposomes. The liposomes were obtained by a simple, organic solvent-free procedure involving the sonication of a phospholipid (P90G) and *S. aethiopicum* peel extract dispersed in water. To evaluate the effect of incorporating the extract into the vesicles, empty liposomes (i.e., without extract) were also prepared and analyzed (Table 4).

The light scattering results, as reported in Table 4, showed that the empty liposomes were approximately 80 nm in diameter, homogeneously dispersed (P.I. < 0.3), and negatively charged. The loading of the extract did not affect the analyzed parameters (i.e., average size, P.I., and zeta potential), as shown in Table 4. Hence, the incorporation of the extract did not interfere with the arrangement of the phospholipid during liposome formation.

### 3.5. Effect of S. aethiopicum Peel Extract on Cell Viability and Intracellular ROS

A cell line that is widely used to test the in vitro toxicity of a plant extract is the human hepatoma cell line HepG2, as it preserves many of the specialized characteristics of normal human hepatocytes [34]. For this reason, HepG2 cells were selected to evaluate the toxicity and the activity of the Lucanian *S. aethiopicum* peel extract. Specifically, to select the appropriate extract concentrations for evaluating peel antioxidant activity, non-cytotoxic concentrations were first evaluated between five different extract concentrations (400–200–100–100–50–25 µg/mL). The test used to evaluate the cytotoxicity was the MTT assay, and its principle is based on the assumption that mitochondrial activity is constant for viable cells leading to an increase or decrease of viable cells proportionally with the mitochondrial activity. The latter is reflected in the tetrazolium salt MTT conversion into formazan crystals; thus, the decrease or increase in viable cells may be detected by formazan concentration measurement [35]. In the case of *S. aethiopicum* peel extract, no cytotoxic effect was observed after 24 and 48 h, and the lowest doses (50 and 25 μg/mL) enhanced cell proliferation by about 30–60% compared to untreated cells (CTRL) (Figure 5).

The absence of extract cytotoxicity confirmed previous data by Akanitapichat et al. [34]; hence, with the exception of the highest concentration (400 µg/mL), the other concentrations were used to evaluate eggplant peel antioxidant activity. Specifically, the antioxidant activity of *S. aethiopicum* peel extract was evaluated in HepG2 cells exposed to *tert*-butylhydroperoxide (*t*-BuOOH). *t*-BuOOH is an organic hydroperoxidant that may be metabolized to free radical intermediates, which in turn start lipid peroxidation, form covalent bonds with cellular molecules, and affect cell integrity, leading to cell injury and death. Toxicity induced by *t*-BuOOH in HepG2 cells is being increasingly used as a model for investigating the cytoprotection of natural antioxidants. As shown in Figure 6, *t*-BuOOH (CTRL+ *t*-BuOOH) increased intracellular ROS, reporting a double fluorescence value compared to untreated cells (CTRL). *S. aethiopicum* peel extract counteracted ROS generation induced by *t*-BuOOH, restoring basal conditions with no statistical differences compared to acetyl-l-cysteine (NAC), a known antioxidant. These data demonstrate the protection of *S. aethiopicum* peel extract against ROS production, in agreement with a previous investigation on the fruit of different African eggplant. Even in these cases, eggplant fruit extract showed a hepatoprotective effect against *t*-BuOOH-induced cytotoxicity [34]. However, this study tested only two concentrations (50–100 µg/mL), while our investigation demonstrated that concentrations lower than 50 µg/mL effectively protected HepG2 cells against oxidative stress as well. Furthermore, in an attempt to improve the performance of the peel extract, the latter was tested when incorporated into liposomes. HepG2 cells were pretreated with *S. aethiopicum* peel extract liposomes for 24 h at the same doses used for raw extract (200–100–50–10–5–1 μg/mL). After 24 h, cells were stressed with t-BuOOH, and antioxidant activity was analyzed. The formulation improved the biological activity of the extract already at low doses (10–1 μg/mL), with fluorescence intensities lower than those of the raw extract (Figure 6). The structure and composition of cell membranes are important factors to consider when administering a biologically active compound, and liposomes may improve its bioavailability, thanks to the nature of their structure. Liposomes are, indeed, nanosized vesicles with one or more phospholipid bilayers that facilitate the penetration of their cargo through the cell membrane, improving their absorption and bioactivity [36]. This is what was observed in this study, since the extract incorporated into liposomes was active at lower concentrations than those required for the raw extract. In support of this conclusion, there are the findings of previous studies, in which the incorporation of extracts from other plant species enhanced their performance in vitro [12,15].

### 3.6. Effect of S. aethiopicum Peel Extract on Antioxidant Defense Markers

Biological systems include a number of antioxidant enzymes involved in the reduction of ROS. Phenols have been reported to increase the expression of body defense enzymes like catalase (CAT), glutathione peroxidase (GPx), and superoxide dismutase (SOD) [15,37]. For this reason, it was decided to test the effect of the Lucanian *S. aethiopicum* peel extract on the expression of different antioxidant enzymes. HepG2 cell line was treated with different doses (200–100 μg/mL) of *S. aethiopicum* peel extract for 24 h, and qRT-PCR was performed to evaluate molecular pathways involved in antioxidant activity (Figure 7).

The extract up-regulated the expression of ATP-binding cassette transporter G2 (ABCG2), NADPH-quinone oxidase (NQO1), CAT, and nuclear factor erythroid 2-related factor 2 (Nrf-2). ABCG2 is an important enzyme involved in maintaining cells’ redox homeostasis, since a deficiency in this protein results in a ROS generation increase. The expression of ABCG2 is highly regulated by hypoxia/oxidative-sensitive transcription factors, such as peroxisome proliferator-activated receptor gamma (PPAR-γ), hypoxia-inducible factor (HIF-1, HIF-2α), and Nrf-2 [38]. Specifically, Nrf-2 is a short-lived protein that can regulate the expression of several cytoprotective genes responsible for xenobiotic metabolism and anti-inflammatory and antioxidant responses. Expression proteins up-regulated by Nrf2 are heme oxygenase-1 (HO-1), SOD1, CAT, and enzymes involved in glutathione metabolism [39]. Hence it is possible to hypothesize that the peel extract of *S. aethiopicum,* cultivated in Basilicata Region, might act on ABCG2 and CAT expression, directly or indirectly, by enhancing Nrf2 expression. However, the peel extract decreased the expression of glutathione peroxidase (GPx-1), and no statistically significant effect on superoxide dismutase (SOD-2) expression was shown.

The obtained results may be related to the high content in flavanones and flavanols, since the most abundant flavanone, naringenin (**10**), was previously demonstrated to enhance the expression of NQO1, CAT, and Nrf2 [40,41]. Similarly, quercetin-3-*O*-rutinoside, or rutin (**6**), the second most abundant compound found in the Lucanian *S. aethiopicum* peel, previously showed an increase in endogenous antioxidant enzymatic activities [42,43].

## 4. Conclusions

The present study evaluated the phytochemical composition and the biological activity of Italian *Solanum aethiopicum* L. growing in the Basilicata Region for the first time. Among the fruit tissues tested, peel extract demonstrated the best antioxidant activity spectrophotometrically and in vitro using HepG2 cell line. The extract protected cells from oxidative stress and enhanced the expression of several antioxidant enzymes such as ABCG2, CAT, and NQO1, presumably via the Nrf2 pathway. Furthermore, it was seen that the incorporation of the extract into liposome increased its antioxidant activity, representing an excellent strategy to improve its specialized metabolites bioavailability in cells, and thus, its activity. Therefore, these results supported and promoted the association between nanocarriers and natural antioxidants to develop health-promoting systems.

## Figures and Tables

**Figure 1 pharmaceutics-14-01168-f001:**
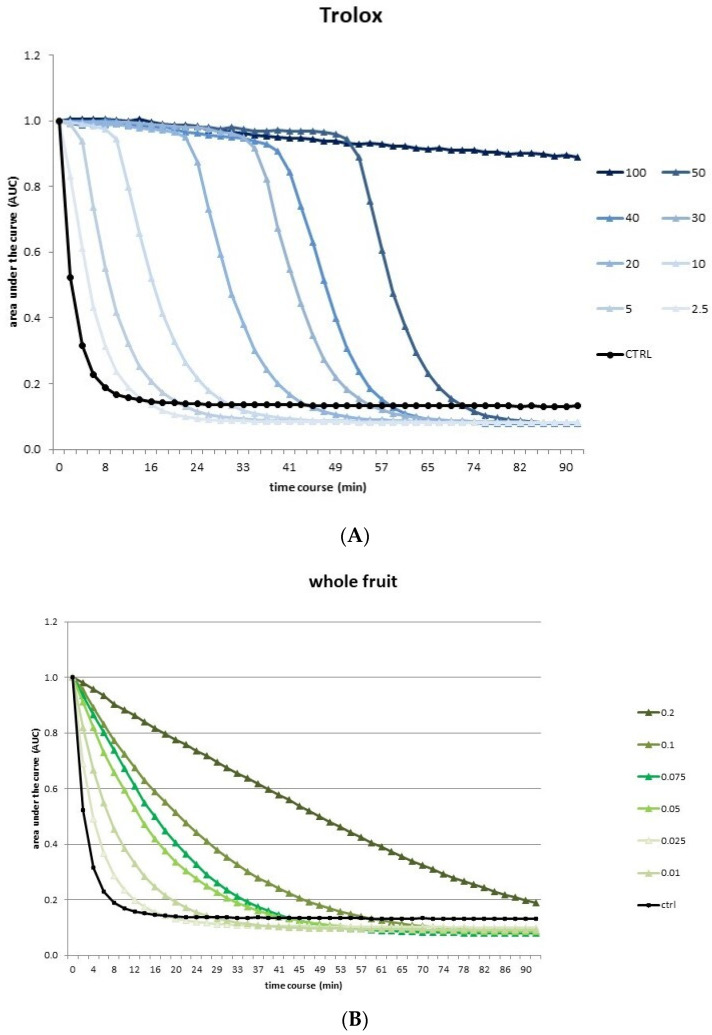
Oxygen radical absorbance capacity (ORAC) assay for different concentrations (2.5–100 µM) of Trolox used as a standard (**A**) and (0.01–0.2 mg/mL) of whole fruit (**B**), pulp (**C**) and peel (**D**). Changes in the fluorescence intensity of fluorescein were monitored for 90 min.

**Figure 2 pharmaceutics-14-01168-f002:**
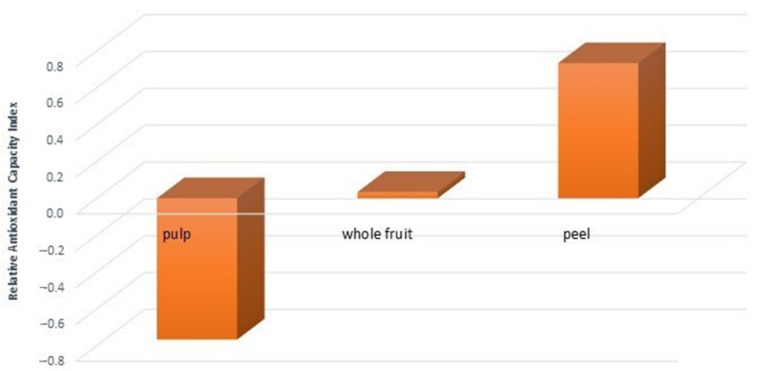
Relative Antioxidant Capacity Index (RACI) values were obtained comparing Total Phenolic Content (TPC), 2,2-diphenyl-1-picrylhydrazyl (DPPH), Ferric Reducing Antioxidant Power (FRAP), and Oxygen Radical Absorbance Capacity (ORAC) results.

**Figure 3 pharmaceutics-14-01168-f003:**
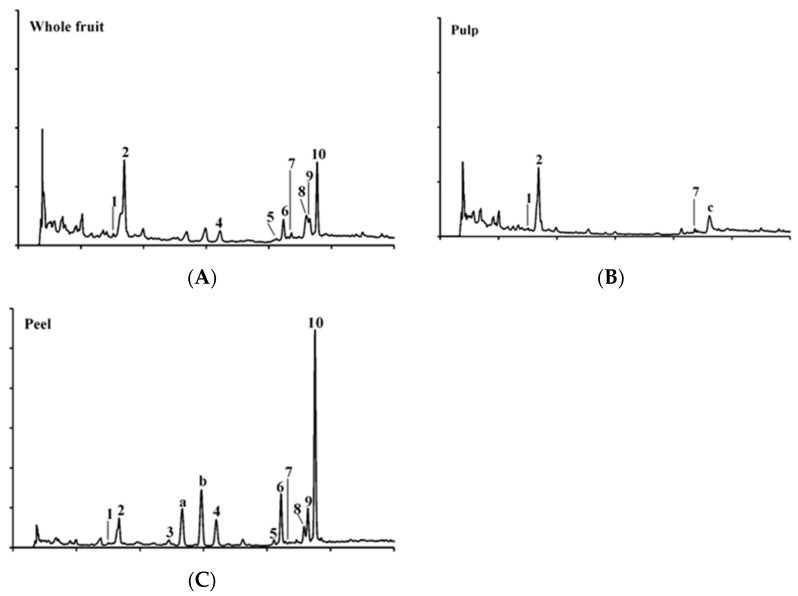
HPLC-DAD phenolic profile of ethanolic extracts of the Lucanian *S. aethiopicum* fruit tissues: (**A**) whole fruit, (**B**) pulp, and (**C**) peel. Detection at 280 nm. (**1**) 4-*O*-caffeoylquinic acid, (**2**) 5-*O*-caffeoylquinic acid, (**3**) eriodictyol-7-*O*-glucoside, (**4**) naringenin-7-*O*-glucoside, (**5**) eriodictyol, (**6**) quercetin-3-*O*-rutinoside, (**7**) 4,5-di-*O*-caffeoylquinic acid, (**8**) kaempferol-3-*O*-glucoside, (**9**) kaempferol-3-*O*-rutinoside, (**10**) naringenin, (**a**, **b** and **c**) unknown flavanones.

**Figure 4 pharmaceutics-14-01168-f004:**
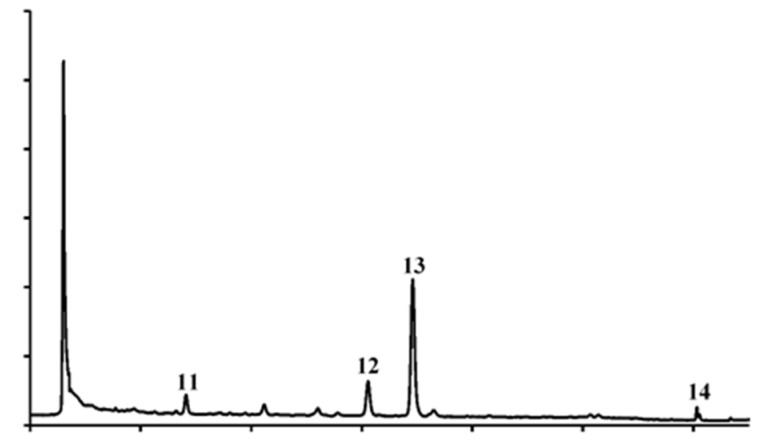
HPLC-DAD carotenoid profile of *S. aethiopicum* peel ethanolic extract. Detection at 450 nm. (**11**) Lutein, (**12**) α-carotene, (**13**) *β*-carotente, (**14**) lycopene.

**Figure 5 pharmaceutics-14-01168-f005:**
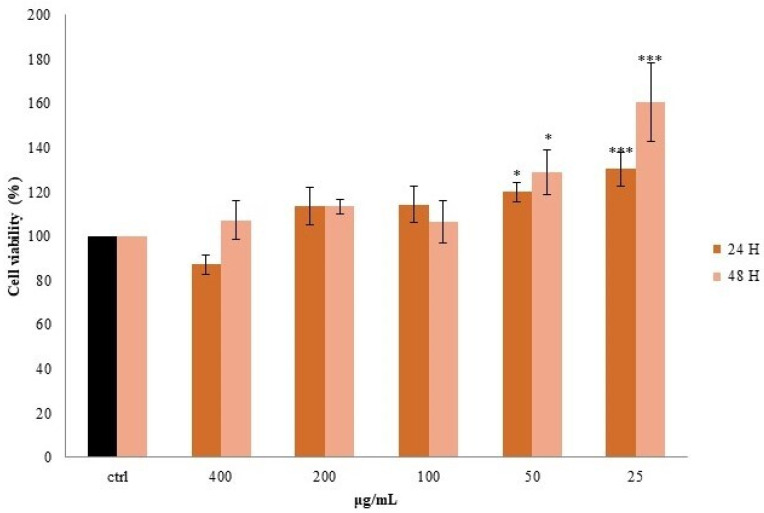
Cell viability, evaluated by MTT assay, of HepG2 cells treated for 24 and 48 h with different concentrations of *S. aethiopicum* peel extract. Data are expressed as the mean ± SD of three independent experiments (*n* = 3). Data are expressed as the mean ± SD of three independent experiments (*n* = 3). * *p* < 0.05, *** *p* < 0.001, ns, not statistically different.

**Figure 6 pharmaceutics-14-01168-f006:**
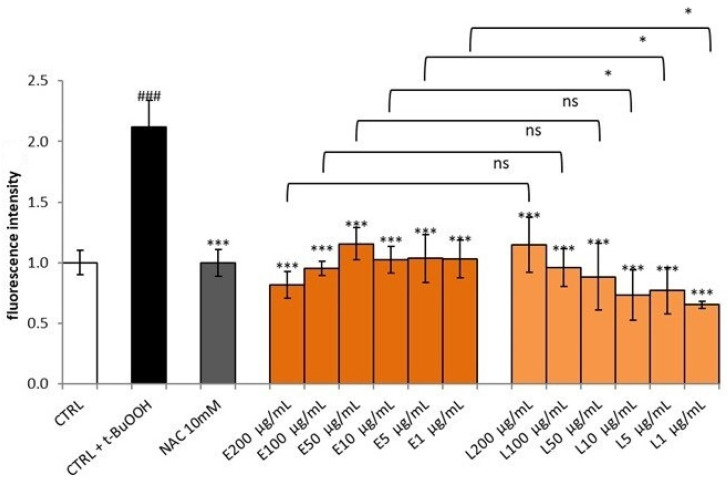
Effects of *S. aethiopicum* extract (E) and extract-loaded liposomes (L) on t-BuOOH-induced intracellular reactive oxygen species (ROS) generation in HepG2 cells. Cells were pretreated with the extract or extract-loaded liposomes at different concentrations (200–1 μg/mL) for 24 h and then incubated with 5 mM *t*-BuOOH for 1 h. Data are expressed as the mean ± SD of three independent experiments (*n* = 3). ### *p* < 0.001 vs. CTRL, *** *p* < 0.001 vs. *t*-BuOOH treated cells, * *p* < 0.05, ns, not statistically different.

**Figure 7 pharmaceutics-14-01168-f007:**
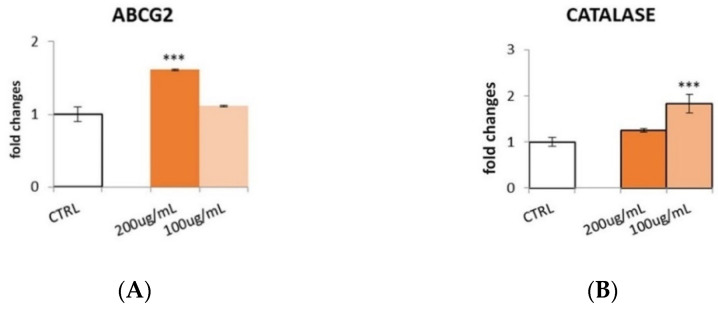
Effect of *S. aethiopicum* peel extract (200 and 100 μg/mL) on the gene expression of (**A**) ATP-binding cassette transporter G2 (ABCG2), (**B**) catalase (CAT), (**C**) glutathione peroxidase (GPx-1), (**D**) NADPH-quinone oxidase (NQO1), (**E**) Nuclear factor erythroid 2-related factor 2 (Nrf-2), (**F**) superoxide dismutase (SOD-2) analyzed by real-time qPCR and normalized with the housekeeping gene, β-actin, in HepG2 cell line. Data are expressed as fold changes, normalized with respect to the untreated control. Data are expressed as mean ± SD of three independent experiments (*n* = 3). ** *p* < 0.01, *** *p* < 0.001 vs. control (CTRL).

**Table 1 pharmaceutics-14-01168-t001:** Extraction yield, total phenolic content (TPC), and antioxidant activity of *Solanum*
*aethiopicum* extracts.

	Extraction Yield (%)	TPC ^1^ (mg GAE/g)	DPPH ^2^ (mg TE/g)	ORAC ^3^ (μmol TE/kg)	FRAP ^4^ (mg TE/g)
**PEEL**	5.82	20.94 ± 0.83 a	16.96 ± 0.57 a	4477.22 ± 330.64 a	21.52 ± 0.58 a
**PULP**	4.72	6.38 ± 0.19 c	4.73 ± 0.14 c	700.37 ± 125.56 c	6.30 ± 0.12 c
**WHOLE** **FRUIT**	5.65	9.41 ± 0.17 b	8.03 ± 0.16 c	1565.31 ± 374.65 b	9.80 ± 0.58 b

^1^ TPC: total phenolic content expressed as milligrams of gallic acid per grams of extract; ^2^ DPPH: 2,2-diphenyl-1-picrylhydrazyl expressed as milligrams of Trolox equivalents per gram; ^3^ ORAC: oxygen radical absorbance capacity assay expressed as micromoles of Trolox equivalents per kg; ^4^ FRAP: ferric reducing antioxidant power expressed as milligrams of Trolox equivalents per grams of extract; Experiments were carried out in triplicate, and data were reported as mean ± SD. Significant differences (*p* < 0.05) are highlighted with different letters (a, b and c).

**Table 2 pharmaceutics-14-01168-t002:** Phenolic content of S. *aethiopicum* fruit tissues (µg/g dry extract) ^1^.

Peak	Compound	Peel	Pulp	Whole Fruit
**1**	4-*O*-Caffeoylquinic acid	–	Nq	56.23 ± 1.33
**2**	5-*O*-Caffeoylquinic acid	1639.82 ± 117.50	1278.22 ± 129.33	1722.19 ± 36.66
**3**	Eriodictyol-7-*O*-glucoside	221.21 ± 12.18	–	–
**4**	Naringenin-7-*O*-glucoside	1270.99 ± 115.80	–	206.86 ± 7.74
**5**	Eriodictyol	106.26 ± 4.868	–	Nq
**6**	Quercetin-3-*O*-rutinoside	4807.21 ± 543.00	–	639.15 ± 29.77
**7**	4,5-di-*O*-Caffeoylquinic acid	Nq	53.22 ± 3.18	91.37 ± 3.77
**8**	Kaempferol-3-*O*-glucoside	712.85 ± 75.17	–	83.58 ± 4.23
**9**	Kaempferol-3-*O*-rutinoside	2320.18 ± 245.93	–	265.78 ± 10.38
**10**	Naringenin	5047.89 ± 509.23	–	558.78 ± 31.56
	TOTAL	16126.40 ± 1623.48	1331.44 ± 132.50	3623.55 ± 106.19

^1^ Results are expressed as the mean ± standard deviation of three determinations. “nq”—not quantified. “–” not found.

**Table 3 pharmaceutics-14-01168-t003:** Carotenoids content of *S. aethiopicum* fruit tissues (µg/g dry extract) ^1^.

Peak	Compound	Peel	Pulp	Whole Fruit
**11**	Lutein	10.84 ± 0.80	–	Nq
**12**	*α*-Carotene	13.61 ± 0.56	–	Nq
**13**	*β*-Carotene	933.80 ± 1.83	–	Nq
**14**	Lycopene	95.06 ± 1.83	–	–
	**TOTAL**	**1053.30 ± 45.16**	–	

^1^ Results are expressed as the mean ± standard deviation of three determinations. “nq”—not quantified. “–” not found.

**Table 4 pharmaceutics-14-01168-t004:** Characterization of empty liposomes and *S. aethiopicum* extract-loaded liposomes.

	Mean Diameter (nm)	P.I.	Zeta Potential (mV)
Empty liposomes	80 ± 6.4	0.27 ± 0.01	−16 ± 3.6
*S. aethiopicum* extract liposomes	77 ± 6.7	0.26 ± 0.03	−20 ± 4.3

Each value represents the mean ± SD, *n* = 10.

## Data Availability

Not applicable.

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
