# Peer review of "New Insight on the Bioactivity of Solanum aethiopicum Linn. Growing in Basilicata Region (Italy): Phytochemical Characterization, Liposomal Incorporation, and Antioxidant Effects"

_pharmaceutics, 2022, doi:10.3390/pharmaceutics14061168_

Round 1
Reviewer 1 Report
The manuscript “New insight on Solanum aethiopicum Linn cv Rotonda extract’ bioactivity: Phytochemical Characterization, Liposomal Incorporation, and Antioxidant effects" presents a novel and well-organized research work on the characterization and bioactivities of S. aethiopicum extracts. In particular, the incorporation of these extracts’ nanoscale liposomes showed very interesting findings on antioxidant properties. Besides, the study is well designed, and the discussion well founded. However, minor modifications are required for the manuscript to be considered for publication in Pharmaceutics, which are suggested in the checklist below.
General observations:
Include references, for example, line 54-65.
Include the brand (city, country) of all laboratory equipment.
Avoid using possessives in the first person and personal pronouns; for example: we, our, us, etc.
Use a comma before the final item in a list of three or more items.
Line 39. prevension of
Line 125. Include storage temperature.
Line 128. 75 µL, concentration?
Line 173. Separate from2 5
Line 249. Please confirm: 200-1 µg/mL
Line 314. All extracts showed a good antioxidant activity.. What is the consideration to define good antioxidant activity?
Line 318. Include reference in square bracket. Nwanna et al. [ ]
Line 347. Please check quantities, they are different than in Table 1.
Tables
Please use same sequence in all Tables (Peel, pulp, whole fruit).
Figures
Improves the quality of all figures.
Figure 1. Change “H” for h
To avoid confusion, it is recommended to separate Figure 5.
Author Response
The manuscript “New insight on Solanum aethiopicum Linn cv Rotonda extract’ bioactivity: Phytochemical Characterization, Liposomal Incorporation, and Antioxidant effects" presents a novel and well-organized research work on the characterization and bioactivities of S. aethiopicum extracts. In particular, the incorporation of these extracts’ nanoscale liposomes showed very interesting findings on antioxidant properties. Besides, the study is well designed, and the discussion well founded. However, minor modifications are required for the manuscript to be considered for publication in Pharmaceutics, which are suggested in the checklist below.
General observations:
Include references, for example, line 54-65.
Include the brand (city, country) of all laboratory equipment.
Avoid using possessives in the first person and personal pronouns; for example: we, our, us, etc.
Use a comma before the final item in a list of three or more items.
Line 39. prevension of
Line 125. Include storage temperature.
Line 128. 75 µL, concentration?
Line 173. Separate from2 5
Line 249. Please confirm: 200-1 µg/mL
Line 314. All extracts showed a good antioxidant activity.. What is the consideration to define good antioxidant activity?
Line 318. Include reference in square bracket. Nwanna et al. [ ]
Line 347. Please check quantities, they are different than in Table 1.
Answer: We thank the reviewer for the recommendations, all suggestions have been included in the text. Referring to “Line 318. Include reference in square bracket. Nwanna et al. [ ]” the numbered reference has been inserted in bracket; Nwanna et al. has been included as subject and author of the study. For “Line 249. Please confirm: 200-1 µg/mL” we confirmed that it is correct, while for “Include the brand (city, country) of all laboratory equipment”, the information is in the text.
Tables
Please use same sequence in all Tables (Peel, pulp, whole fruit).
Answer: all tables have been corrected
Figures
Improves the quality of all figures.
Figure 1. Change “H” for h
To avoid confusion, it is recommended to separate Figure 5.
Answer: all suggestions have been included in the text
Reviewer 2 Report
In this paper, author investigated a Solanum aethiopicum extract bioactivity: phytochemical characterization, liposomal incorporation, and antioxidant effects. I found this study is very useful with correctly interpreted results. The
authors did well on the discussion section.
Author Response
Thank you for your positive comments
Reviewer 3 Report
Scarlett eggplant (Solanum aethiopicum L.) was investigated with respect to its the antioxidant activity and the phytochemical composition. The whole fruit, peel, and pulp were subjected to EtOH exhaustive maceration extraction. The HPLC-DAD analysis of the extracts revealed the presence of ten phenolic compounds, including hydroxycinnamic acids, flavanones, and flavonols and three carotenoids such as one xanthophyll and three carotenes. The peel extract was the most promising, the most active and the richest in specialized metabolites and therefore it was tested on HepG2 cell lines and incorporated into liposomes. The nanoincorporation enhanced peel extract’s antioxidant activity, resulting in a reduction of the concentration used. Furthermore, the extract improved the expression of endogenous antioxidants, such as ABCG2, CAT, and NQO1, presumably through the Nrf2 pathway. The manuscript is interesting; the research was performed with adequate methods. I found only minor remarks that are listed further.
Minor remarks:
Lines 89-90 : ”Hence, this study 89 aimed to investigate for the first time the phytochemical composition of S. aethiopicum…” . However later in discussion (lines 392-393) it is mentioned that only 5-O-caffeoylquinic acid and quercetin-3-O-rutinoside were recently reported in S. aethiopicum fruit aqueous and methanolic extracts [2,21,26]”. Therefore, there is need to explain more the novelty of performed research and to mention these previously performed researches in the introduction part as well. The novelty should be pointed out with respect to [2,21,26].
Why the authors have chosen conventional extraction with EtOH and not with some other new innovative techniques? Why EtOH was chosen as the extraction solvent?
Lines 350-352: “This is in line with the nature of phenolic compounds, since they have been regarded as potent antioxidants in vitro and have been shown to be more effective antioxidants than vitamin C, vitamin E, and carotenoids” – There is need to cite appropriate reference to confirm this statement about phenolic compounds as more potent antioxidants than vitamin C, vitamin E, and carotenoids.
Lines 397-398: “…flavanone-like UV-vis spectra, they were labelled as unknown flavanones (a–c)…”. Why the authors have not made attempt to tentatively identified these compounds, e. g. by using HPLC-MS?
Can identified carotenoids contribute to the observed antioxidant activity? If that is possible than it should be included in the discussion part of the manuscript.
Author Response
Scarlett eggplant (Solanum aethiopicum L.) was investigated with respect to its the antioxidant activity and the phytochemical composition. The whole fruit, peel, and pulp were subjected to EtOH exhaustive maceration extraction. The HPLC-DAD analysis of the extracts revealed the presence of ten phenolic compounds, including hydroxycinnamic acids, flavanones, and flavonols and three carotenoids such as one xanthophyll and three carotenes. The peel extract was the most promising, the most active and the richest in specialized metabolites and therefore it was tested on HepG2 cell lines and incorporated into liposomes. The nanoincorporation enhanced peel extract’s antioxidant activity, resulting in a reduction of the concentration used. Furthermore, the extract improved the expression of endogenous antioxidants, such as ABCG2, CAT, and NQO1, presumably through the Nrf2 pathway. The manuscript is interesting; the research was performed with adequate methods. I found only minor remarks that are listed further.
Minor remarks:
Lines 89-90 : ”Hence, this study 89 aimed to investigate for the first time the phytochemical composition of S. aethiopicum…” . However later in discussion (lines 392-393) it is mentioned that only 5-O-caffeoylquinic acid and quercetin-3-O-rutinoside were recently reported in S. aethiopicum fruit aqueous and methanolic extracts [2,21,26]”. Therefore, there is need to explain more the novelty of performed research and to mention these previously performed researches in the introduction part as well. The novelty should be pointed out with respect to [2,21,26].
Answer: all citations [2,21,26] referred to the African eggplant. We have now specified it to avoid franteindiments
Why the authors have chosen conventional extraction with EtOH and not with some other new innovative techniques? Why EtOH was chosen as the extraction solvent?
Answer: thanks for the observation additional information has been added in the text, in future studies we will try to use other, more innovative techniques.
Lines 350-352: “This is in line with the nature of phenolic compounds, since they have been regarded as potent antioxidants in vitro and have been shown to be more effective antioxidants than vitamin C, vitamin E, and carotenoids” – There is need to cite appropriate reference to confirm this statement about phenolic compounds as more potent antioxidants than vitamin C, vitamin E, and carotenoids.
Answer: we have added it
Lines 397-398: “…flavanone-like UV-vis spectra, they were labelled as unknown flavanones (a–c)…”. Why the authors have not made attempt to tentatively identified these compounds, e. g. by using HPLC-MS?
Answer: thank you for the suggestion, it has now been inserted in the text this will be done in future studies
Can identified carotenoids contribute to the observed antioxidant activity? If that is possible than it should be included in the discussion part of the manuscript.
Answer: thank you for the suggestion, it has now been inserted in the text